

# Deep learning Binary/Multi classification for music's brainwave entrainment beats

Rowayda A. Sadek[1], Alaa A. Khalifa[1] and Marwa M.A. Elfattah[2]

[1] Department of Information Technology, Faculty of computers and Artificial Intelligence, Helwan University, Cairo, Egypt
[2] Department of Computer Science, Faculty of computers and Artificial Intelligence, Helwan University, Cairo, Egypt

## ABSTRACT

The last two decades have seen the emergence of a brand-new kind of music known as digital brain stimulant, also known as instrumental music or music without lyrics, which mostly comprises entrainment beats. While listening to it has the same ability to affect the brain as taking medication, it also has the risk of having a negative impact or encouraging unwanted behavior. This sparked the interest of a large number of studies in the psychological and physiological effects of music's brainwave entrainment beats on listeners. These studies started to categorize and examine how musical beats affected brainwave entrainment by looking at electroencephalogram (EEG) signals. Although this categorization represents a step forward for the early research efforts, it is constrained by the difficulty of having each musical track and conducting EEG tests on humans exposed to distortion due to noise in order to determine its influence. The work proposed in this article continues to explore this topic but in a novel, simple, accurate, and reliable categorization procedure based on the music signal elements themselves rather than dependent on EEG. VGGish and YAMNET based transfer deep learning models, are tuned to handle a straightforward, accurate real-time detector for the existence of the music beats inside music files with accuracy of 98.5 and 98.4, respectively. Despite the fact that they yield results that are equivalent, the YAMNET model is more suited for use with mobile devices due to its low power consumption and low latency. The article also proposes modified version of VGGish and YAMNET binary classifying models called BW-VGGish and BW-YAMNET respectively. The modification was to turn the binary classification into multi-classification. These multi-classifiers handle the classification of the influence of music beats (five different brain waves) on human brainwave entrainment with average accuracy of 94.5% and 94.5%, respectively. Since there was a lack of datasets addressing this kind of music, two datasets, the Brainwave Entrainment Beats (BWEB) dataset and the Brainwave Music Manipulation (BWMM) dataset, were generated for classification training and testing. The re-testing on a sample of music files that have their impact on brain waves (with their EEG) in an earlier study is done to strengthen the validity of the proposed work and to overcome the potential limitation of utilizing a music dataset that is not proved with its EEG. The success of the suggested models was demonstrated.

Corresponding author
Alaa A. Khalifa,
alaa.abdelaziz@fci.helwan.edu.eg

# INTRODUCTION

Music has become an integral part of our daily lives as we listen to it during our various daily activities. As a result, many types of music have appeared to suit all tastes, including music that incorporates entrainment beats that may affect the brain in an analogous way to how drugs do, while also having a negative impact or encouraging undesirable behaviors (*Jirakittayakorn & Wongsawat, 2017*; *Wahbeh, Calabrese & Zwickey, 2007*). These beats can aid with anxiety, concentration, memory, mood, creativity, pain relief, meditation, cognitive flexibility, and sleep quality. Listening to it, on the other hand, might have side effects, such as increasing feelings of sadness, anxiety, and anger in certain people.

As a result, much research had been done on the psychological and physical impacts of music's brainwave entrainment beats on people using the electroencephalogram (EEG) signals. But these research face the challenge of having each musical track and conducting EEG experiments on humans to identify it, as well as the EEG recordings (*Repovš, 2010*), can be effected by different types of noise such as the electrical interference, electromyographic (EMG), and electrooculogram (EOG). Thus, it is important to detect the existence of the brainwave entrainment beats within music files and to categorize music files according to how they affect brain waves using the music signal features themselves rather than the EEG.

The bidirectional deep long short term memory (BDLSTM) model (*Sadek, Khalifa & Elfattah, 2021*) is a modified version of the long short term memory (LSTM) model. This model was used for its ability to learn, process, and categorize sequential data; it can also learn long-term relationships between time steps of data. It is successful to detect the presence of entrainment beats in music files with an accuracy of 90.02%, which serves as an example of why deep learning techniques are the most effective models for this problem. The BDLSTM model was hampered by the tiny amount of data that it had to work with, but this problem can be solved by applying the transfer learning methodology (*Bali & Tyagi, 2020*; *Gemmeke et al., 2017*).

As a result, our contributions to this article are:

- The detection of the existence of brainwave entrainment beats within music files with higher accuracy than the BDLSTM model using the VGGish and YAMNET models; binary classification
- The categorization of music beats files according to how they affect brain waves based on five different brain waves using the BW-VGGish and BW-YAMNET models; Multi-classification
- A Brainwave Musical Manipulation (BWMM) dataset was provided.
- Carry out a real testing verification process.

This work's remaining sections are organized as follows: The Literature Review section contains some information on earlier research on this topic. The structure of the dataset is discussed in the Dataset section. The architecture of the classification models is discussed in the Proposed Models section. The experiment and its results are discussed in the

Experimental Results section. The model's verification is discussed in the Proposed Models Verification section, which is followed by sections on conclusion and references.

## LITERATURE REVIEW

Based on our knowledge of all the research that was previously conducted on the effect of entrainment beats on brain waves, it has been carried out in two directions. In the first direction, the effect of these beats and the possibility of using them from a medical point of view were studied, while in the second direction, it was studied how to analyze and classify the effect of these beats on brain waves using electroencephalogram (EEG) waves and artificial intelligence techniques.

**Medical view:** *Becher et al. (2014)* presented a study that examined the modification of the electroencephalogram (EEG) power and phase synchronization by stimulation through sound with monaural and binaural beats in epilepsy patients before surgery. The results showed that phase synchronization can be modulated using audio stimulation, and it offered a non-surgical model for the modulation of intracranial EEG syncing that could have medicinal applications.

*Beauchene et al. (2016)* presented a study that determined the response accuracy and cortical network topology, which were measured by EEG recordings using different acoustic stimulation. The results showed that listening to 15 Hz binaural beats increased the response accuracy and modified the strengths of the cortical networks.

*Shumov et al. (2017)* presented a study that provided a comparative analysis of the time to fall asleep using three types of files: binaural beats (BB) with pink noise, a sound with monaural beats (MB), and a sound without any beat (imitations, IM). The results showed that listening to the soundtrack with BBs has an average time to fall asleep less than the time needed when listening to the soundtrack with MBs or IM.

*Engelbregt et al. (2019)* presented a study that examined the effects on attention and working memory in high and low emotional participants when listening to the binaural beats compared with the monaural beats using the stimulation process. A flanker task to measure attention and a Klingberg task to measure working memory were performed, and the results showed that for the attention, the performance speed was higher in the MB and BB conditions relative to white noise (WN), while in the Klingberg task, no significant differences were found under the WN, MB, and BB conditions.

*Ross & Lopez (2020)* presented a paper that shows the possibility of improving the training outcome in an attentional blink (AB) task using binaural beat stimulation. Magnetoencephalography (MEG) recordings show that there is a high improvement of gamma waves during 40 Hz binaural beats stimulation and a smaller improvement with 16 Hz binaural beats.

*Yusim & Grigaitis (2020)* presented a study that had the goal of finding the effects of binaural beats on brain activity. They compared the effects of the binaural beats with those of monaural beats, the results showed that binaural beats are weakly entrained in the cortex in comparison with monaural beat stimuli.

*Engelbregt et al. (2021)* presented a study that showed the 40 Hz monaural beats (MB) and binaural beats (BB) effects on attention and electroencephalogram (EEG). The

result showed that the binaural beats improved attention on the Flanker task, but this improvement did not appear in the electroencephalography.

**Technical view:** Many studies have been conducted on EEG-based recognition, where *Tai & Lin (2018)* presented a study that compares between the EEG signals of the brainwaves influenced by classical music and heavy metal music. In this study, nine people wear headsets and listen to two music files, which are Sonata for Two Pianos in D major, K.448'' and ''Napalm Death Procrastination On The Empty Vessel'' for 10 min then their brainwaves were collected. Based on the collected brainwaves, the results confirmed that Listening to the classical music led to a drastic improvement in mental performance.

*Johnson & Durrant (2021)* investigated the neural mechanisms of brain-wave music on sleep quality. In this study, 33 participants listened to brainwave music for 20 min before bedtime for 6 days, and their EEGs were collected. According to the results, slow wave sleep brain-wave music has a positive effect on sleep. Furthermore, an increase in the functional connectivity (FC) between the left frontal and parietal lobes and a drop in the delta band's power spectral density may be responsible for enhanced sleep quality.

According to our understanding and former literature assessment, no other experimenter worked on how to detect the presence or absence of brainwave entrainment beats inside audio recordings in real time. All the research that was conducted was to prove the effect of entrainment beats on different brain waves using EEGs.

First, the bidirectional deep long short-term memory (BDLSTM) model (*Sadek, Khalifa & Elfattah, 2021*) was used to detect the presence of brainwave entrainment rhythms in music files using music features. This model was based on the LSTM model with gaining average accuracy 90.2%. However, the problem with the dataset's small size can be solved by using the transfer learning method, according to a study by *Bali & Tyagi (2020)*, which showed that VGG16 transfer learning is a powerful technique where the knowledge gained from the larger dataset is transferred to the new dataset. It also showed that it is preferable to use transfer learning techniques rather than CNNs as it's hard for the CNNs to be trained using small datasets. Also, *Barman et al. (2019)* performed a comparison between transfer learning ResNet50 and normal learning, and it was explained that transfer learning requires less data as compared to a model built from scratch, requires less computation power, and does not require a long time, whereas the pre-trained model does most of the heavy work, like recognizing major patterns, and does not require heavy GPU's. Even working with huge data can be done on a decent CPU without a GPU.

When looking at previous studies on transfer learning techniques, some studies have been found, including (*Haq, 2021*) offers research that uses deep learning algorithms to classify music genres using the AudioSet dataset. This study compares the Vgg 19 transfer learning model, which had a 75 percent accuracy rate, with the Vgg 19 fine-tuning, which had a 78 percent accuracy rate, and the Vgg feed-forward baseline, which had an 81% accuracy rate, while *Shi et al. (2019)* present a study that uses the VGGish-BiGRU model, which is a VGGish network paired with a bidirectional gated recurrent unit neural network, and the dataset of 1,152 samples to distinguish lung sounds. This model was chosen because the information contained in lung sounds is too complicated to be displayed using standard characteristics, and the temporal features of lung sounds cannot be recovered using a

convolutional neural network. The settings of the VGGish layers are fixed, while the BiGRU network's parameters are fine-tuned. In the recognition of asthma accuracy, this model performs better than state-of-the-art algorithms like CNN.

In order to get around the dataset's small size, transfer learning was utilized. We tested a number of models before settling on the best one for identifying entrainment beats in music files. The most accurate models were then adjusted for use in the process of categorizing music files based on their impact on brain waves.

## DATASETS

Due to the lack of previous research that studied the entrainment beats using the music features themselves, there is a lack of datasets that include music files with brainwave entrainment beats. As a result, the brainwave entrainment beats (BWEB) dataset and the Brainwave Musical Manipulation (BWMM) dataset were created to be used in the experiments that we conducted during our study.

The music files for the BWEB and BWMM datasets were taken from SoundCloud and YouTube audio files. Audience comments and number of views guided the selection of audio files for download (*Gemmeke et al., 2017*). Each track lasts 30 s and is sampled at a rate of 44,100 samples per second in 32 bits.

### Brainwave entrainment beats (BWEB) dataset

The BWEB dataset (*Khalifa, Sadek & Elfattah, 2022*), is a 300-track self-gathered collection organized into two classes: brainwave entrainment music and non-entrainment music. Each class has an aggregate of 150 tracks, bearing in mind that each class contains all the different sorts that make up these classes. The brainwave entrainment class contains music files with three different entrainment beats, which are monaural beats, binaural beats, and isochronic beats, and the non-entertainment class contains different music files that have no entrainment beats. This set of files was put together to diversify the dataset and contain all the available possibilities for each class. Also, by adding an equal number of files to each kind, a balance has been maintained between the various classes.

### Brainwave musical manipulation (BWMM) dataset

The BWMM dataset (*Sadek, Khalifa & Elfattah, 2021*), is a 240-track self-collected collection organized into six classes which are, alpha, beta, delta, theta, gamma, and no entrainment. There are 40 tracks overall in each category. There are different types of files in each category. The tracks in the No Entrainment category were put up using audio files from numerous music genres as well as ambient sounds, whereas the three types of brainwave entrainment monaural, binaural, and isochronic are included in the other five categories. This variation of files was created to make the dataset more diversified and to include all the available options for each category. Also, by adding an approximately equal number of files to each kind, a balance has been maintained between the various categories. For example, the alpha category contains 40 files from different three categories which are 13 files for monaural, 14 for binaural, and 13 for isochronic.

Samples for the dataset were taken from audio files on the servers of SoundCloud and YouTube. Based on audience feedback, the audio samples were picked for download. Each

track lasts for 30 s and is sampled at a rate of 44,100 samples per second at a bit depth of 32. Samples were taken from each audio file in a systematic method, covering the beginning, middle, and end of each audio file separately.

## PROPOSED MODELS

The proposed models that we reached have the ability to classify music files based on the presence of brainwave entrainment beats inside them, in addition to the ability to classify music files based on their effects on brain waves (five different brain waves). To get to these models, we have gone through two phases. In the first phase, we have gone through many models that had drawbacks that led us to use another model to overcome these drawbacks until we reached our proposed models. All these models have been trained and tested with the Brainwave Entrainment Beats (BWEB) dataset in the process of detecting the existence of entrainment beats inside music files. Then we went to the second phase, in which the models that achieved the highest accuracy were developed in the first phase for use in the process of classifying music files based on their effect on brain waves.

The models that we presented in the first phase are: AS-BDLSTM, VGGish, and YAMNET. All these models were trained and tested using the Brainwave Entrainment Beats (BWEB) dataset (*Khalifa, Sadek & Elfattah, 2022*). Firstly, we developed the AS-BDLSTM model using the transfer learning methodology by training it first on the AudioSet dataset and then using it with the target dataset, which improved the results but required a long training time and a lot of memory. This is what prompted us to consider employing the VGGish and YAMNET models, which were developed by Google.

While in the second phase, we developed the BW-VGGish and BW-YAMNET models from the VGGish and YAMNET models by training them on the Brainwave Entrainment Beats (BWEB) dataset in addition to their pretraining on the AudioSet dataset, then using them with the target dataset, which is BrainWave Musical Manipulation (BWMM). Because the BWEB dataset contains data that is similar to the target dataset, BW-VGGish and BW-YAMNET gained more experience when training on it. This experience makes BW-VGGish and BW-YAMNET more accurate in the categorization of music files based on their effects on brain waves.

There are six different effects that brain waves can have.

· **Delta** (People in this state enter a state of deep sleep)

Beats range: 0.5–4 Hz

· **Theta** (People in this state enter a first state of sleep and feel drowsiness)

Beats range: 4–8 Hz

· **Alpha** (People in this state should be relaxed but alert)

Beats range: 8–13 Hz

· **Beta** (People in this state should be highly alert and focus)

Beats range: 13–30 Hz

· **Gamma** (People in this state should be able to recall memories)

Beats range: >30 Hz

· **No effect** (No entrainment beats)

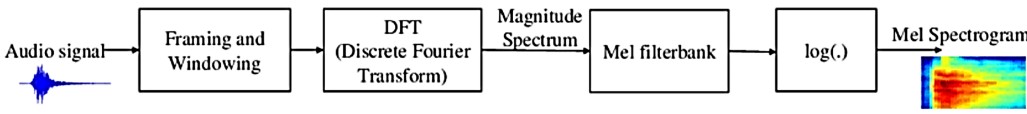

**Figure 1** Block diagram of Mel spectrogram of an audio signal.

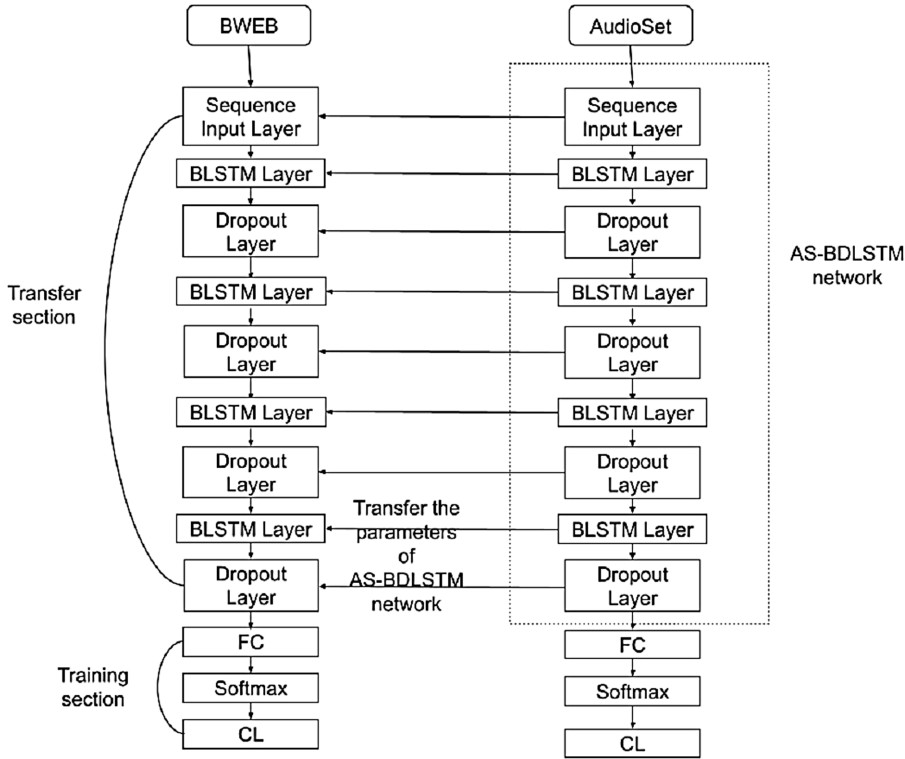

**Figure 2** AS-BDLSTM model architecture.

All the proposed models (VGGish, YAMNET, BW-VGGish, BW-YAMNET) were trained on the log mel spectrogram of the data, which made it necessary to pre-process both the BWEB and BWMM datasets with the following steps:

- Convert audio signals to the model's learned sampling rate.
- Obtain the log mel spectrogram as shown in Fig. 1, which is divided into 96 frames with 48-frame overlap.

## AS-BDLSTM

As shown in Fig. 2, the AS-BDLSTM approach was constructed by first training the BDLSTM approach on the AudioSet dataset, then retraining the output approach on the target dataset, which was a BWEB dataset. The retraining process is called the transfer learning process, which transfers the first training parameters to the target network. AudioSet and BWEB datasets were pre-processed, and the Mel-FrequencyCepstral Coefficients (MFCC) and Gammatone Cepstral Coefficients (GTCC) features were

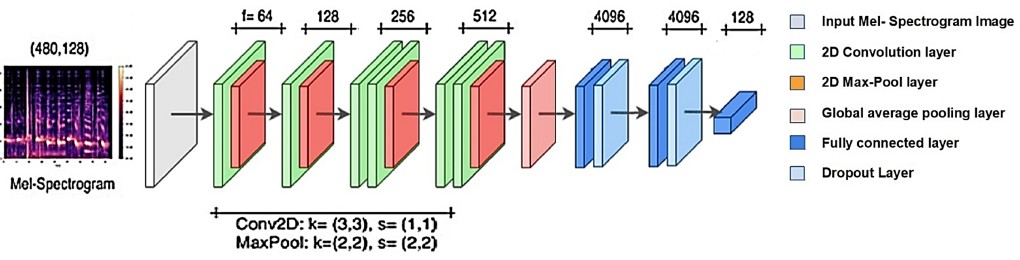

**Figure 3** VGGish general model scheme.

extracted as feature matrix in the form of sequence data, which was sent into the network as a sequence input through the sequence input layer.

## VGGish model

The VGGish model is a modification of the VGG model (*Hershey et al., 2017*) and it is appropriate to be used with tiny datasets (*Bali & Tyagi, 2020*; *Gemmeke et al., 2017*). It is pre-trained on the AudioSet dataset, which is a large-scale labeled audio dataset that Google opened in 2017. It consists of 24 layers. Six convolutional layers and three fully connected layers are among the nine layers with learnable weights, as shown in Fig. 3.

The VGGish model is a ready-made model; when using it, we use the same layers of the model with the removal of the last layer to match the size of the number of the out classes and adding a group of fully connected, softmax and classification layers. Then we train it on the target data set with an input size of 96 × 64, as shown in Fig. 4. During the training process, the training parameters are transformed from the VGGish network to the target network, which enhances the performance result.

## YAMNET model

YAMNET (Yet Another Mobile Network) is a pre-trained neural network that uses the MobileNetV1 depthwise-separable convolution architecture. MobileNet is a mobile and embedded visual convolutional neural network. They are based on a condensed architecture that builds thin deep neural networks using depthwise separable convolutions (*Harjoseputro, Yuda & Danukusumo, 2020*). It is characterized by low latency and low power consumption. The scheme of a generic YAMNET deep neural network for classification tasks is represented in Fig. 5, as the input features go through the input layer and then the input features are gone through multiple convolutions. Finally, the output of the convolution layers go through the fully connected layers, which send the outs to the output layer that is out the results.

YAMNET is made up of 86 layers. There are 28 layers with weights that may be learned: There are 27 convolutional layers and one fully connected layer. It is a pre-built model trained on the AudioSet dataset from Google. With the exception of deleting the last layer to match the number of out classes and adding a set of entirely connected, softmax, and classification layers, we use the same model layers as previously. Then, with a 96 × 64

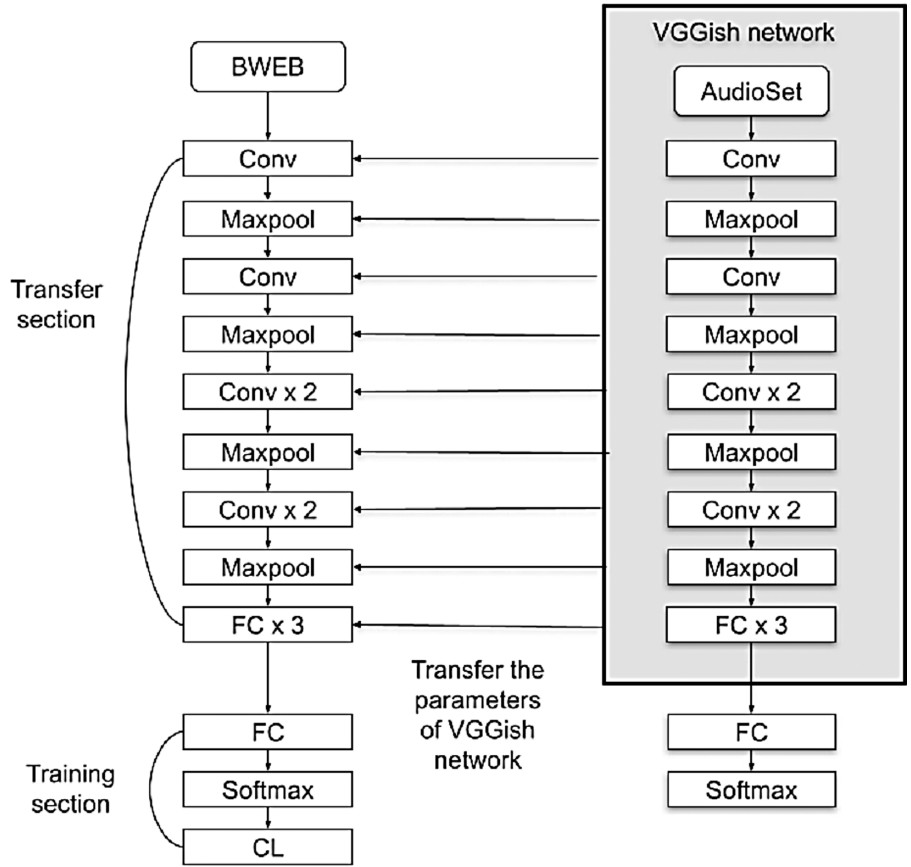

**Figure 4  VGGish model architecture.**

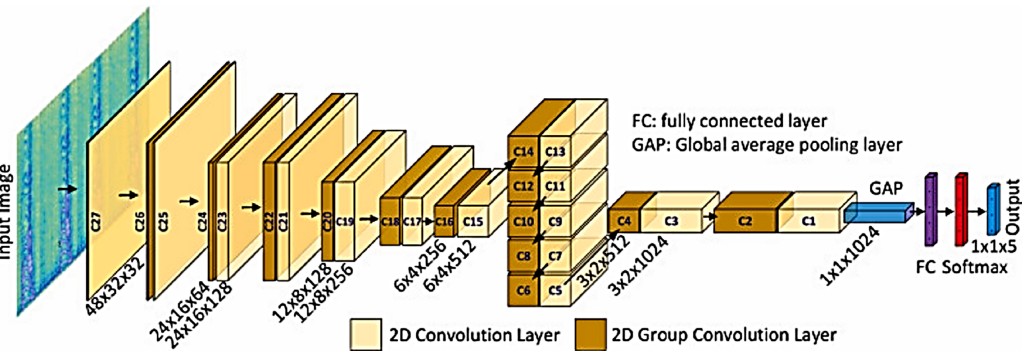

**Figure 5  YAMNET model architecture.**

input size, we train it on the target data set. Throughout the training process, the training parameters are switched from the YAMNET network to the target network, which enhances the performance outcome.

### BW-VGGish model

BW-VGGish model is a developed version of the VGGish model. It is pre-trained on the BWEB dataset in addition to its pre-training on the AudioSet dataset. Unlike the VGGish model which pretrained on only the AudioSet dataset. The training on the BWEB dataset has been done on to make the model gain more experience in classifying this type of music that contains brainwave entrainment beats inside it.

The BW-VGGish model architecture is shown in Fig. 6, where the VGGish model parameters transferred to the BW-VGGish network while training of the VGGish model on BWEB dataset, then the model parameters transferred to the target network while the training of the BW-VGGish model on a BWMM dataset.

### BW-YAMNET model

BW-YAMNET model is a developed version of the YAMNET model. It is pre-trained on the BWEB dataset in addition to its pre-training on the audioset dataset. Unlike the YAMNET model, which is trained on only the AudioSet dataset, the training on a BWEB dataset has been done to give the model more experience in classifying this type of music that contains brainwave entertainment beats inside it.

In the BW-YAMNET the YAMNET model parameters are transferred to the BW-YAMNET network while training the YAMNET model on BWEB dataset, then the model parameters are transferred to the target network while training the BW-YAMNET model on a BWMM dataset.

## EXPERIMENTAL RESULTS

To test the generalization ability of the model, the experiments were run three times, and the mean of the results was calculated as the average precision. In each experiment, the k-fold cross-validation (*Haq, 2021*; *Shi et al., 2019*) with $k = 5$ was used, where the data was separated into five subgroups at random and the ratio of samples in each subset was the same for each category. Four subsets were taken as the training data in turn, and one subset as the testing data. From each training subset, 20% of the data was taken for validation. All the experiments in this article have been done on windows 10.1 with a hardware core of I7, 16 GB of RAM using MATLAB R2020b (*The MathWorks, Inc., 2020*).

The hyper-parameters that were utilized to train the models are:

- The learning rate equals 0.0001
- Mini-batch size equals 32
- Optimizer equals adam
- Training using one GPU

After that, we verified the effectiveness of the proposed models using data that was outside the data set used in the training process. We tested music files that were used and

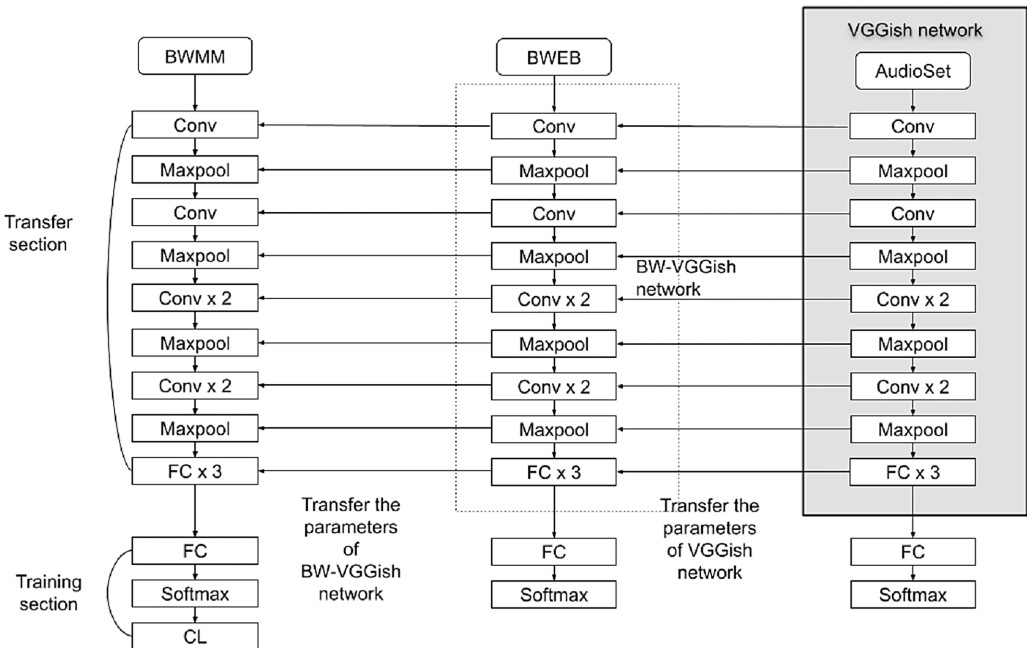

**Figure 6** BW-VGGish model architecture.

classified in previous research using EEG, then compared the results of their classification with our results, as will be discussed in the Proposed Models Verification section.

## Performance evaluation metrics of model

Accuracy, recall, precision, F-score, and the confusion matrix are all used to evaluate the model's performance.

$$Accuracy = \frac{(TN + TP)}{(TP + TN + FP + FN)} \tag{1}$$

$$Precision = \frac{TP}{(TP + FP)} \tag{2}$$

$$Sensitivity/Recall = \frac{TP}{(TP + FN)} \tag{3}$$

$$F - score = 2 * \frac{Precision * Recall}{Precision + Recall} \tag{4}$$

where TP indicates how many times the model properly predicts the positive category, TN indicates how many times the model has accurately predicted the negative category, FP indicates the number of times the model has predicted the positive category incorrectly, and FN indicates how many times the model has predicted the negative category erroneously.

**Table 1  Model comparison between VGGish, YAMNET, and BDLSTM.**

| Model | Precision (%) | Recall (%) | F-score (%) | Accuracy (%) |
|---|---|---|---|---|
| VGGish | 99.12 | 99.33 | 99.22 | 98.5 |
| YAMNET | 100 | 98% | 98.98 | 98.4 |
| AS-BDLSTM | 92 | 90.67 | 91.33 | 91.1 |
| BDLSTM | 89 | 92 | 90.47 | 90.2 |
| Unidirectional - LSTM | 86 | 90.6 | 88.24 | 88.4 |

**Table 2  Model comparison between VGGish and BW-VGGish with the BWMM dataset.**

| Model | Precision (%) | Recall (%) | F-score (%) | Accuracy (%) |
|---|---|---|---|---|
| VGGish | 93.9 | 93.75 | 93.82 | 93.3 |
| BW-VGGish | 95 | 95 | 95 | 94.6 |

**Table 3  Model comparison between YAMNET and BW-YAMNET with the BWMM dataset.**

| Model | Precision (%) | Recall (%) | Accuracy (%) |
|---|---|---|---|
| YAMNET | 93.66 | 93.83 | 93.3 |
| BW-YAMNET | 94.66 | 94.66 | 94.6 |

## RESULTS DISCUSSION

Table 1 shows the performance metrics for the unidirectional LSTM, BDLSTM, AS-BDLSTM, VGGish, and YAMNET models (*Sadek, Khalifa & Elfattah, 2021*). The AS-BDLSTM model achieved higher percentages for precision, and accuracy with 92%, and 91.1%, respectively when compared with the BDLSTM model. On the other hand, it has a lower recall than the BDLSTM model, but this can be bypassed as we care more about the precision value. However, our greatest concern is the correct detection of the entrainment beats inside the music files and we can disregard mistakenly labeling a music file as containing entrainment beats. Although the AS-BDLSTM model improved efficiency, it required a long time for training and a large memory space, which made us move on to thinking about using the VGGish and YAMNET models.

The VGGish and YAMNET models achieved higher percentages for precision, recall, and accuracy when compared with the BDLSTM model. VGGish and YAMNET models provide similar performance metrics, but the YAMNET model is recommended for usage with mobile apps because of the nature of its structure. It is a light - parameter model that needs less power, which is a requirement for mobile applications.

Tables 2 and 3 show the performance metrics for the VGGish, BW-VGGish, YAMNE, and BW-YAMNET models when used with the BWMM dataset in the classification of the music files depending on their effect on the brain waves.

The performance of the models can be visualized using the confusion matrix. Figures 7 and 8 represent the confusion matrix for the transfer learning of the VGGish and YAMNET

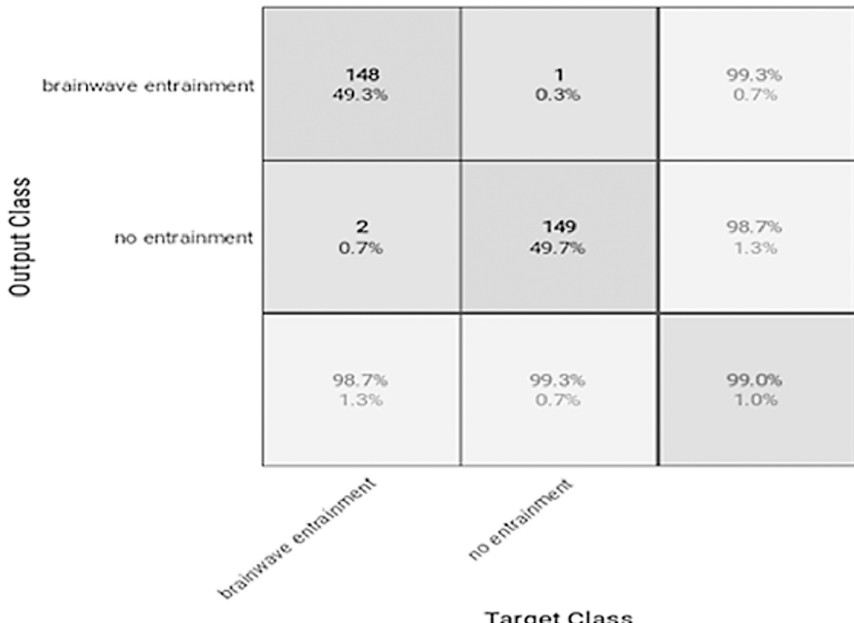

**Figure 7  Confusion matrix of the VGGish model with the BWEB dataset.**

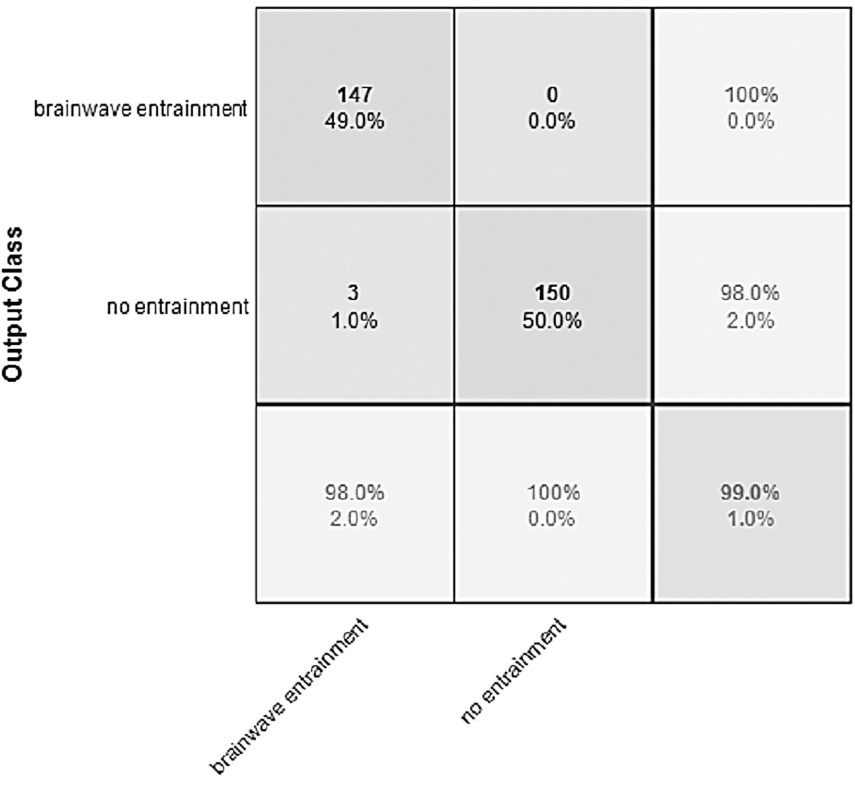

**Figure 8  Confusion matrix of the YAMNET model with the BWEB dataset.**

**Figure 9** Confusion matrix of the VGGish model with the BWMM dataset.

models with the BWEB dataset, while Figs. 9–12 represent the VGGish, BW-VGGish, YAMNET, and BW-YAMNET models when used with the BWMM dataset.

## PROPOSED MODELS VERIFICATION

In this section, we will review the experiments and results that we carried out to verify the effectiveness of the proposed models using data that is outside the data set used in the training process. We tested music files that were used and classified in previous research, then compared the results of their classification with our results. In this experiment, we downloaded two music files, which are "Sonata for Two Pianos in Dmajor, K.448" for classical music and "Napalm Death Procrastination on The Empty Vessel" for metal music. These music files were used by *Tai & Lin (2018)* to study the effect of classical and metal music on brain waves. In *Tai & Lin (2018)*, nine volunteers listened to the two music files for 10 min. Then their brain waves were collected for 8 min only by removing the data for the first and last minute. When entering these music files into our models, we found that "Sonata for Two Pianos in D major, K.448" affected the delta waves, while "Napalm Death

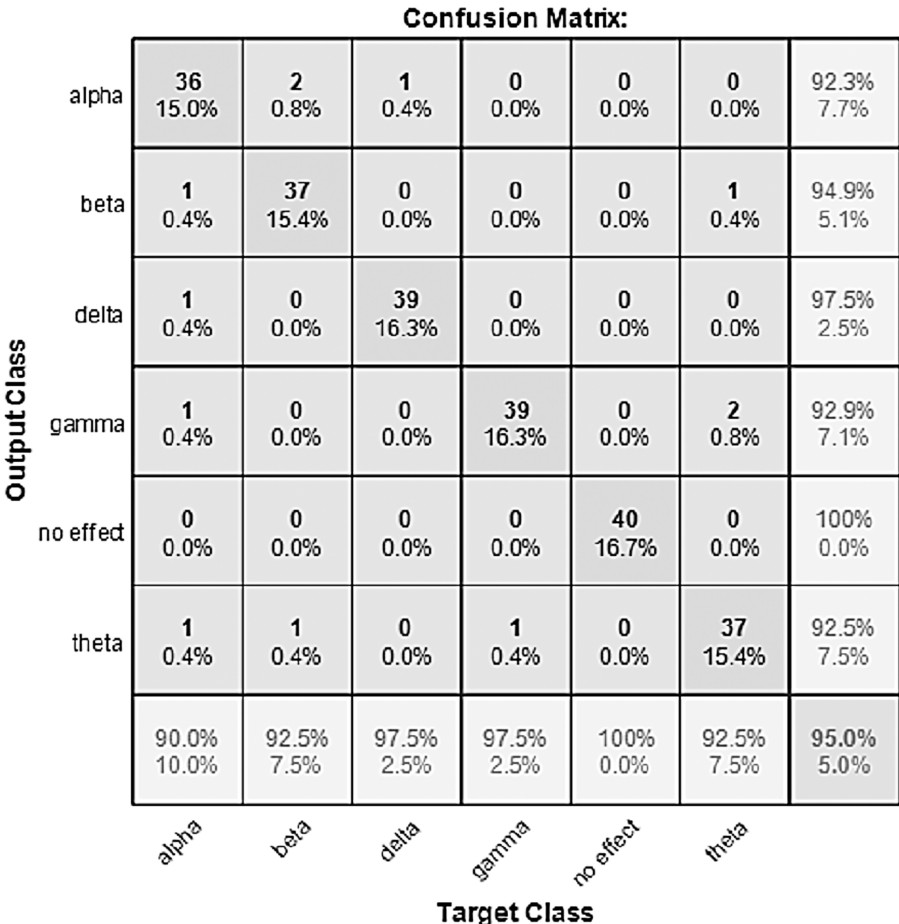

**Figure 10  Confusion Matrix of the BW-VGGish Model with the BWMM Dataset.**

Procrastination on The Empty Vessel'' affected the beta waves. And this result is the same that has been proven in *Tai & Lin (2018)* as shown in Table 4.

## CONCLUSIONS

Music has become a significant part of people's lives as a result of the growing quantity of songs available and the widespread usage of the Internet by people of all ages. They listen to music at various times throughout the day and engage in diverse activities. A new kind of music has emerged in recent years that combines brainwave entrainment beats that can alter human moods and induce seizures in epileptic sufferers. This kind of music has grown rapidly, as evidenced by the fact that there are multiple sites that sell these audio files for a price, in addition to their free availability on YouTube and the SoundCloud server.

In this article, we present models to be employed in two key categorization processes. The VGGish and YAMNET models are proposed to be employed in the first classification process, which is the categorization of music files based on the presence of entrainment

**Figure 11  Confusion matrix of the YAMNET model with the BWMM dataset.**

beats within them. We suggested the BW-VGGish and BW-YAMNET models for the second classification, which is the categorization of music files based on their impact on brain waves.

The VGGish and YAMNET models overcome the BDLSTM model with an accuracy of 98.5% and 98.4%, respectively. BW-VGGish and BW-YAMNET models are modifications of the VGGish and YAMNET models. This modification happened by pre-training them on the BWEB dataset in addition to the AudioSet dataset to give them additional experience with music files containing brainwave entrainment beats. This knowledge has given them a leg up on the competition when it comes to categorizing music files based on their effect on brain waves. Both the BW-VGGish and BW-YAMNET models achieved an accuracy of 94.6%.

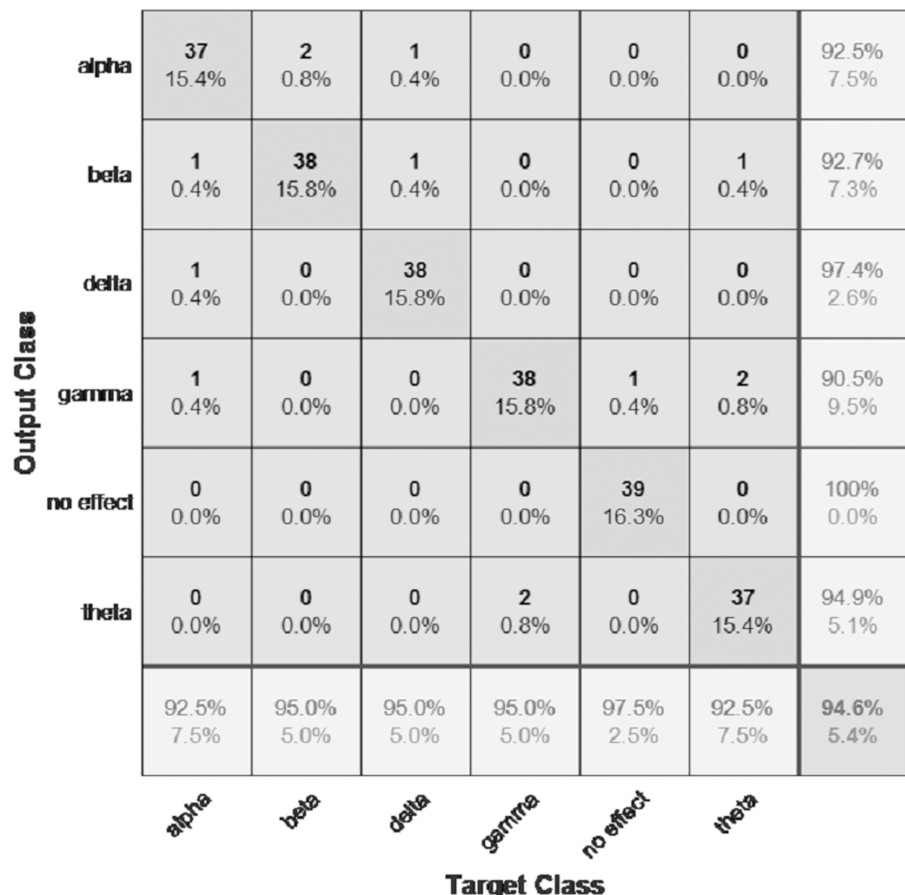

**Figure 12** Confusion matrix of the BW-YAMNET model with the BWMM dataset.

**Table 4** A comparison between the results of our models and previous research.

| Music file | BW-VGGish /BW-YAMNET | Previous research |
|---|---|---|
| Sonata for Two Pianos in D major, K.448 | Delta waves | Delta waves |
| Napalm Death Procrastination On The Empty Vessel | Beta waves | Beta waves |

### Funding
The authors received no funding for this work.

### Competing Interests
Rowayda A. Sadek is an Academic Editor for PeerJ.

### Author Contributions
- Rowayda A. Sadek conceived and designed the experiments, authored or reviewed drafts of the article, and approved the final draft.

- Alaa A. Khalifa performed the experiments, performed the computation work, prepared figures and/or tables, and approved the final draft.
- Marwa M.A. Elfattah analyzed the data, authored or reviewed drafts of the article, and approved the final draft.

### Data Availability

The data is available at GitHub and Zenodo:

- https://github.com/AlaaKhalif/BWEB.git

- AlaaKhalif. (2023). AlaaKhalif/BWEB: Deep learning Binary /Multi classification for music's brainwave entrainment beats (BWMM). Zenodo. https://doi.org/10.5281/zenodo.8341595

### Supplemental Information

Supplemental information for this article can be found online at http://dx.doi.org/10.7717/peerj-cs.1642#supplemental-information.

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
