# Peer review of "Deep learning Binary/Multi classification for music’s brainwave entrainment beats"

_PeerJ Computer Science, doi:10.7717/peerj-cs.1642_

## Round 0.1 · original submission · Major Revisions

Both reviewers find merit in the paper and recommended a major revision. Authors are required to revise the manuscript and address all the comments and suggestions of the reviewers. One of the reviewers suggested modifying the title and rewriting the contents of the manuscript. The revised manuscript will be subjected to re-review. Good luck.

**Language Note:** The review process has identified that the English language must be improved. PeerJ can provide language editing services - please contact us at copyediting@peerj.com for pricing (be sure to provide your manuscript number and title). Alternatively, you should make your own arrangements to improve the language quality and provide details in your response letter. – PeerJ Staff

Reviewer 1 ·

Basic reporting

1. The manuscript is about the effect of music beats on the brain using deep learning approach.

2. It is well-written but needs a major revision.

3. Abstract needs to be revised.

4. Introduction section should be expanded. There should be sub-section like ‘Our Contribution’, key work summary in point-wise.

5. All figures are looking not clear.

Experimental design

1. Why F-score is not available in all results?
2. Ablation study is missing.
3. Comparison with neural network baseline models are missing like CNN, LSTM, etc.. Only, compared with existing work.
4. Results in terms of Precision and Recall are very high and looking unrealistic. Justify it.
5. Please share the GitHub code link in the paper.

Validity of the findings

The main goal of this manuscript is to highlight the effect of music beats on the brain using deep learning which needs to addressed properly in terms of evaluation results and comparative analysis. Also, present form of the manuscript needs a major revision and precise explanation.

·

Basic reporting

The English language and grammar need improvement.

The literature review may need additional literature please see comment 5.

The article structure needs improvement, the Methodology and experimental protocol is sparse, keep items missing.

Hypotheses missing.

Ket terms not defined.

Experimental design

1. The term “as well as its distortion from noise exposure” is not very clear, as if the subjects are listening to music for brain enhancement they will not try it in a noisy environment. Secondly, nowadays the use of earphones/headphones is very common, hence the external source of noise can be avoided. However f the authors are talking about general music listening in concerts or shopping malls, then it is a different scenario. Its purpose is not brain entrainment. Hence the above statement needs clarification. And noise is nowhere discussed in the rest of the manuscript.
2. EEG Signals---all abbreviations should be explained at the first appearance. What is BDLSM (line 62) stand for?
3. It will be good if the authors give a brief overview of what the BDLSM model is for, before going into further details in the introduction section.
4. Here it is interesting to note that not only the brain entrainment beats may have an effect on the human brain (identified by EEG) but the lyrics of the music in which it is embedded may also have different effects, hence if let’s say brain entrainment beat AA is embedded in music X it may have a slightly different effect if it is embedded in music Y. So how these effects can be identified if the model is only identifying the entrainment beats AA.
5. It will be good if, in the literature review, the authors should include an overview of the literature that has compared the effect of binaural beats with music and established the superiority of one over the other in terms of brain entrainment and vice versa,
6. Line 84, “Electromagnetic (EEG) waves”
7. Line 101, when listening to the soundtrack with MBs or IM. What is MBs and IM?
8. Line 106, “MB and BB conditions” Whats BB?
9. Line 113, should be “presented”
10. Line 119, how it was established that attention was improved?
11. Line 131, FC?
12. Line 138, authors should first elaborate on their work and its results, after that may explain the deep model. As it is not very clear from the brief statement.
13. It is recommended that at the end of the literature review, the authors establish their hypotheses. It looks like the authors want to identify the presence of entrainment beats in music audio, however the title of the paper is misleading which shows refer to the “effect of the music beats on the brain.
14. In order to achieve the objective of understanding the effects of the music beats on the brain, a brain physiological qualitative/quantitative analysis is needed. However till now at the end of the literature review, it looks like the authors are combining the entrainment waves and music and try to identify the presence of beats in the music using Deep learning.
15. Line 173, what kind of comments were used to select the music, as the audience comments/likeness cannot be used for music classification in terms of being useful for human cognition etc, for the purpose entrainment beats are used.
16. Line 183, 194, “different music files that have no effect on the brain waves” This statement may not be correct as all music has some effect on the human brain and emotions.
17. So authors may first identify the effect of individual music and beats and then combine it with the beats, and study the effect of 3, only then the effect of that piece of audio can be established.
18. Line 199, proposed work, if the authors are trying to classify different music then it makes sense, but when they say classify music based on the effect on the brain then it does not make sense, as they have not studied the effect of the music types on the brain after they combine the beats with different music selected only based on user reviews without conducting any experiments on studying their individual effect on the brain.
19. Hence this work can be considered a classification problem for identifying embedded beats in music only.
20. Line 323 to 331, the information regarding the proposed model verification experiment has not been mentioned earlier in the manuscript, under experiment design, hence the experimental protocol is not very clear. Earlier during methodology the authors were just discussing model training. It is recommended to clearly identify the experiment protocol at the start of the methodology.
21. The English language , grammar, and spellings need a thorough check.

Validity of the findings

The findings may be valid, but they are not in line with the title of the work. Please see the comments in section 2.

Additional comments

Needs a rewrite.

---

## Round 0.2 · Minor Revisions

Dear Authors,

Some of the comments of the reviewers were not properly addressed. I invite you to revise the manuscript and address the suggested concerns, and submit a clear response to the comments.

Reviewer 1 ·

Basic reporting

Authors have addressed all comments. Paper can be accepted in the present form.

Experimental design

Authors have addressed all comments. Paper can be accepted in the present form.

Validity of the findings

Authors have addressed all comments. Paper can be accepted in the present form.

Additional comments

Accept

·

Basic reporting

The authors have revised the manuscript but i am having problems in looking at the review comments. For some reason, the authors have not included their reply in the response file and have mentioned the line nos. where changes have been made, but i believe the line numbers mentioned are not correct so it is difficult to follow the revision.

Experimental design

I requested to explain the protocol/method for dataset generation as well as data collection, It will be good to explain in words as well as draw a flow chart or block diagram but it was not provided. For example how long is the data segments, how EEG data is collected etc. This has not been discussed in spite of my comments. What equipment was used?

The procedure says it has seen the effect of 5 brain ways, but its not clear.

How BWMM dataset was created


Procedure for real-time testing and verification process.

LIne 259-260 reads "The ability to classify music based on the effect on brainwaves..."

Again it has not been mentioned in the paper how EEG was measured, what hardware was used and what experimental method was adopted.

Validity of the findings

Seems valid but cannot verify

Additional comments

Line 69 replace lake with lack.

---

## Round 0.3 · accepted · Accept

The authors have addressed all the comments of the reviewers. The manuscript may be accepted for publication in its current form.